# Transcriptome analysis of Kentucky bluegrass subject to drought and ethephon treatment

Jiahang Zhang[1], Yanan Gao[2], Lixin Xu[1]\*, Liebao Han[1]\*

1 College of Grassland Science, Beijing Forestry University, Beijing, People's Republic of China, 2 College of Grassland Science and Technology, China Agricultural University, Beijing, People's Republic of China

\* lixinxu@bjfu.edu.cn (LX); hanliebao@163.com (LH)

**Data Availability Statement:** All relevant data are within the paper and its Supporting information files.

**Funding:** This work was funded by Fundamental Research Funds for the Central Universities (grant

## Abstract

Kentucky bluegrass *(Poa pratensis L.)* is an excellent cool-season turfgrass utilized widely in Northern China. However, turf quality of Kentucky bluegrass declines significantly due to drought. Ethephon seeds-soaking treatment has been proved to effectively improve the drought tolerance of Kentucky bluegrass seedlings. In order to investigate the effect of ethephon leaf-spraying method on drought tolerance of Kentucky bluegrass and understand the underlying mechanism, Kentucky bluegrass plants sprayed with and without ethephon are subjected to either drought or well watered treatments. The relative water content and malondialdehyde conent were measured. Meanwhile, samples were sequenced through Illumina. Results showed that ethephon could improve the drought tolerance of Kentucky bluegrass by elevating relative water content and decreasing malondialdehyde content under drought. Transcriptome analysis showed that 58.43% transcripts (254,331 out of 435,250) were detected as unigenes. A total of 9.69% (24,643 out of 254,331) unigenes were identified as differentially expressed genes in one or more of the pairwise comparisons. Differentially expressed genes due to drought stress with or without ethephon pretreatment showed that ethephon application affected genes associated with plant hormone, signal transduction pathway and plant defense, protein degradation and stabilization, transportation and osmosis, antioxidant system and the glyoxalase pathway, cell wall and cuticular wax, fatty acid unsaturation and photosynthesis. This study provides a theoretical basis for revealing the mechanism for how ethephon regulates drought response and improves drought tolerance of Kentucky bluegrass.

## 1. Introduction

Environmental factors (such as light, temperature, water, soil, etc.) are very important to the growth and survival of plants, because slight changes of these environmental factors may make plants suffer from stress, thus affecting their normal growth and survival [1]. Drought stress is one of the major factors limiting plant growth and crop productivity in many areas [2]. It reduces the turf quality of Kentucky bluegrass by influencing the shoot density, texture, uniformity, color, growth habit and recuperative capacity [3, 4]. The common responses of plants to drought stress include the expression changes of many genes, such as genes related to signal

numbers 2021ZY83) and the National Natural Science Foundation of China (grant numbers 31971770).

**Competing interests:** The authors have declared that no competing interests exist.

transduction, and the transcription and regulation of thousands of functional proteins, which are involved in the molecular regulation of drought resistance [5].

The phytohormone ethylene is a key signaling molecule in plants for regulating multiple developmental processes and stress responses [6, 7]. As an ethylene releasing reagent, ethephon can overcome the disadvantage of inconvenient application of gaseous ethylene and has great potential in practice for various reasons [8, 9]. For example, researches on Maize (*Zea myus*) [10], rice (*Oryza sativa*) [11] and *Arabidopsis thaliana* [12] have revealed that ethephon could improve plants drought tolerance. Few studies focused on the potential of ethephon application in turfgrass species for water saving reasons. Zhang et al. [13] found that ethephon seeds treatment improved drought tolerance of Kentucky bluegrass seedlings by increasing antioxidant enzyme activity and soluble protein content under PEG-induced drought conditions. Han [14] found that specific concentration of ethephon could effectively improve the drought tolerance of Kentucky bluegrass. However, it is still unclear how ethephon affect the response mechanism of Kentucky bluegrass under drought.

At present, a large number of studies have revealed the mechanism for plants drought tolerance through transcriptome sequencing [15–18]. Illumina sequencing technology has been used in the study of turfgrass genome such as *Lolium temulentum* L. [19], orchardgrass (*Dactylis glomerata L.*) [20] and creeping bentgrass (*Agrostis stolonifera*) [21]. For species without genome information, transcriptome sequencing can effectively characterize and identify the biosynthesis pathway of secondary metabolites in plants, reveal the growth, development, physiological adaptability of plants, and explore the gene sequence and expression level [22–24]. Zhang et al. [25] compared the transcriptome of drought resistant and sensitive plants collections of Qinghai wild *Poa pratensis* under drought, and found that genes involved in the starch and sucrose metabolism pathways, and bHLH, AP2/EREPB and $C_2H_2$ zinc finger family transcription factors played important roles in drought tolerance of Kentucky bluegrass. Leng et al. [26] revealed that genes encoding protein kinase, protein phosphatase, genes involved in carbon metabolism and ABA synthesis and transduction are crucial in *Kentucky bluegrass* 'Nuglade' drought defense responses. Gene expression changes on a whole transcriptome level associated with ethephon pre-treatment under drought stress of Kentucky bluegrass have not been well-studied yet.

The objective of this study is to investigate the effect of ethephon on drought tolerance of Kentucky bluegrass and to understand the underlying mechanism by analyzing and identifying genes involved in ethephon mediated drought tolerance improvement.

## 2. Materials and method

### 2.1 Plant materials and treatment

Seeds of Kentucky bluegrass (cv. Nuglade) were from Beijing Top Green Company. All materials were planted in the greenhouse of Turfgrass Reasearch Station of Beijing Forestry University, Bajia nursery, Beijing, China. The plants were grown in plastic pots (diameter: 20 cm, depth: 18 cm) filled with a mixture of peat, vermiculite and perlite (2:1:1). Plants were watered every 2 days to keep the soil moisture conditions at field capacity. Drought stress was imposed by withdrawing water for 13 days until soil moisture drop to 4% (portable time domain reflectometry)(TZS-I, Zhejiang TOP Instrument Co., Ltd, China). Ethephon solution (200 mg/L) was foliar-sprayed 7 days ahead of drought treatment. After 15 days of drought treatment, the upper 3–5 leaves were sampled from well-watered control plants without ethephon application (CK), drought treated plants without ethphon application (Drought), and droughttreated plants with ethephon pre-treatment (ETH_D) for RNA sequencing and real-time PCR analysis (Fig 1).

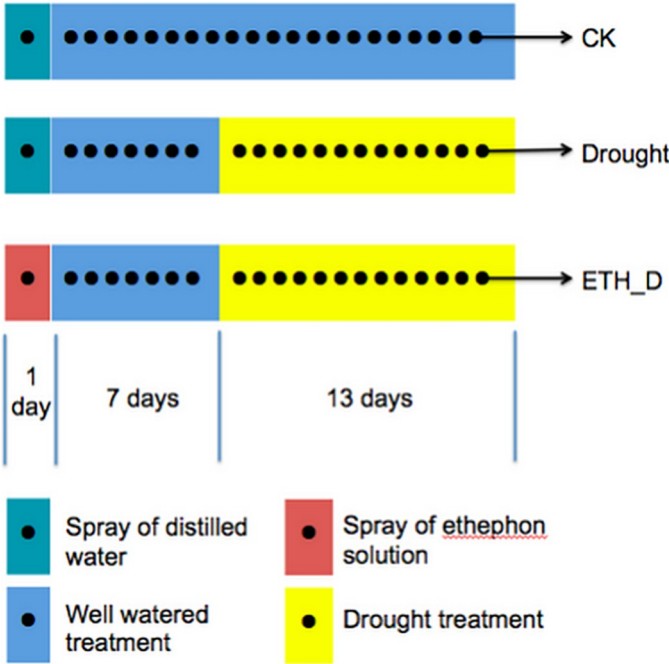

**Fig 1. Schematic overview of the experimental design for well-watered control plants without ethephon application (CK), drought control plants without ehtphon application (Drought), and drought control plants with ethephon pre-treatment (ETH_D).** CK means well-watered control plants without ethephon application. Drought means drought-stressed plants without ethephon application. ETH_D means drought-stressed plants with ethephon pre-treatment.

## 2.2 Relative water content and malondialdehyde content

The relative water content (RWC) of leaves was determined by drying method [27]. The content of malondialdehyde (MDA) ($\mu mol \cdot g^{-1}$) was determined by thiobarbituric acid method [28].

## 2.3 RNA isolation and library preparation

Total RNA was extracted using TRIzol kit (Invitrogen, CA, USA) according to the manufacturer's instructions and was treated with DNaseI. RNA purity was checked using the Nano-Photometer® spectrophotometer (IMPLEN, CA, USA); RNA integrity was assessed using the RNA Nano 6000 Assay Kit of the Agilent Bioanalyzer 2100 system (Agilent Technologies, CA, USA). RNA integrity number for the RNA samples are in the range from 6.3 to 7.2.

The cDNA library was prepared by pooling RNA from the leaf samples of CK, Drought, ETH_D. Three biological replicates for each treatment were used for RNA sequencing and real-time PCR analysis.

A total amount of 3 μg RNA per sample was used as input material for the RNA sample preparations. Sequencing libraries were generated using NEBNext® UltraTM RNA Library Prep Kit for Illumina® (NEB, USA) following manufacturer's recommendations and index codes were added to attribute sequences to each sample. Briefly, mRNA was purified from total RNA using poly-T oligo-attached magnetic beads. Fragmentation was carried out using divalent cations under elevated temperature in NEBNext First Strand Synthesis Reaction Buffer (5X). First strand cDNA was synthesized using random hexamer primer and M-MuLV Reverse Transcriptase (RNase H-). Second strand cDNA synthesis was subsequently

performed using DNA Polymerase I and RNase H. Remaining overhangs were converted into blunt ends via exonuclease/polymerase activities. After adenylation of 3' ends of DNA fragments, NEBNext Adaptor with hairpin loop structure were ligated to prepare for hybridization. In order to select cDNA fragments of preferentially 150~200 bp in length, the library fragments were purified with AMPure XP system (Beckman Coulter, Beverly, USA). Then 3 μl USER Enzyme (NEB, USA) was used with size-selected, adaptor-ligated cDNA at 37°C for 15 min followed by 5 min at 95°C before PCR. Then PCR was performed with Phusion High-Fidelity DNA polymerase, Universal PCR primers and Index (X) Primer. At last, PCR products were purified (AMPure XP system) and library quality was assessed on the Agilent Bioanalyzer 2100 system.

## 2.4 Sequencing, assembly, and annotation

Sequencing, assembly, and annotation were performed by Novogene Bioinformatics Technology Co. Ltd (https://www.novogene.com/). The clustering of the index-coded samples was performed on a cBot Cluster Generation System using TruSeq PE Cluster Kit v3-cBot-HS (Illumia) according to the manufacturer's instructions. After cluster generation, the library preparations were sequenced on an Illumina Hiseq platform and paired-end reads were generated.

The raw reads were sequenced on the Illumina HiSeq™ 4000 platform. After the raw reads containing adapter sequences, reads containing ploy-N ($\geq$10%) and low quality reads had been removed, the clean reads were assembled de novo using the Trinity (http://trinityrnaseq.github.io) as previous described [22]. It was a K-mer length of 25 and a minimum assembly length at 200bp that default parameters were set for fast and efficient transcript assembly. The longest transcript of each gene is used as a representative of the gene, called Unigene, for subsequent analysis. Taxonomic and functional annotation of all spliced sequences was obtained by comparing with seven databases which were the NCBI non-redundant protein sequences (NR) database, NCBI nucleotide sequences (NT) database, protein family (PFAM) database, eukaryotic ortholog groups (KOG) database, Swiss-Prot database, Kyoto Encyclopedia of Genes and Genomes (KEGG) database and Gene Ontology (GO) database. Based on the protein annotation result of NR and PFAM, analysis of the Gene Ontology (GO) term was conducted for functional annotations (E-values<$10^{-6}$). The KAAS software was used to blast the gene sequences in the unigene and the KEGG gene database.

## 2.5 Identification of differentially expressed genes

The transcriptome obtained by Trinity splicing were the reference sequence, and the clean reads of each sample were mapped directly to the reference transcriptome libraries using the RSEM (v1.2.15) software [29] with default parameters. Readcount for each gene was obtained from the mapping results. Differential expression analysis of three treatments was performed using the DESeq R package [30]. The resulting P values were adjusted using the Benjamini and Hochberg's approach for controlling the false discovery rate [31]. Genes with an adjusted P-value <0.05 found by DESeq were assigned as differentially expressed.

GO enrichment analysis of differentially expressed genes (DEGs) was performed by GOseq method [32] based on Wallenius non-central hyper-geometric distribution. The analysis first mapped all the differentially expressed genes to each term of the Gene Ontology database, calculated the number of genes for each term, and then found the significant enrichment in the differentially expressed genes compared to the entire genome background. Simultaneously, up regulated and down regulated genes was performed separately for enrichment analysis in order to better study the function of differential genes.

## 2.6 Validation of differential expression genes by qRT-PCR

Eight differentially expressed genes were randomly selected for qRT-PCR analysis, and high-through put data were validated (the prime pairs of these eight genes were listed in S1 Data). Total RNA was extracted respectively from the leaves of each sample as previous described. Complementary DNA from total RNA was prepared using HiScript® II Q RT SuperMix for qPCR kit (with the gDNA wiper) (Vazyme Biotech Co., Ltd, Nanjing, China) according to the manufacturer's protocol. The primers were designed for qRT-PCR and the Actin gene was used as the internal reference gene. The qRT-PCR was carried out using the Applied Biosystems 7500 real-time PCR system. The relative quantitative data were calculated using the $2^{-\Delta\Delta CT}$ method [33].

## 3. Results and discussion

### 3.1 Relative water content and malondialdehyde content

Relative water content (RWC) can be used as an index to measure the internal water loss and water holding capacity of plants. Higher RWC under drought stressed conditions means better drought tolerance of Kentucky bluegrass [2, 34]. The value of RWC in Kentucky bluegrass leaves decreased significantly by drought. Ethephon pre-treated plants maintained a higher level of RWC under drought stress relative to non-ethephon treated control plants (Fig 2A). Therefore, ethephon could improve the drought tolerance of Kentucky bluegrass. Malondialdehyde (MDA) is a final product of plant cell membrane lipid peroxidation and is widely used as a biomarker of oxidative stress in plants [35]. Under drought stress, lower MDA content is associated with better stress tolerance of turfgrasses [36]. MDA content in Kentucky bluegrass leaves increased significantly by drought, ethephon pre-treatment lowered the level of MDA under drought (Fig 2B). These results together confirmed that ethephon could improve the drought tolerance of Kentucky bluegrass by combining the photos of ethephon and drought treatment (Fig 3).

### 3.2 Sequence assembly

A set of 435,250 transcripts was produced using Trinity. We selected 254,331 sequences (58.43% of the total transcripts) as unigenes, with a mean length of 581 bp and an N50 of 818 bp (see S2–S4 Data for data used to summarize the quality of sequencing, assembly and alignment). The Kentucky bluegrass 254,331 assembled unigenes were queried against seven

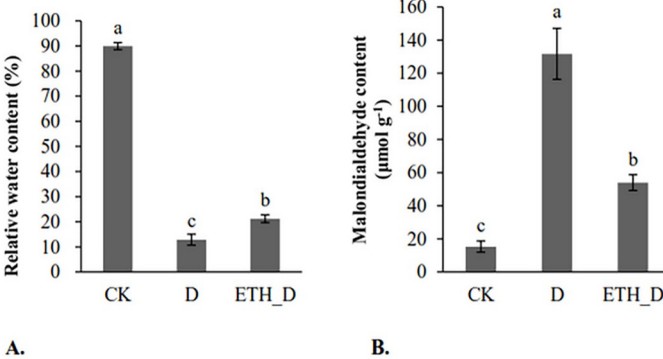

**Fig 2.** A. Relative water content of each samples B. Malondialdehyde content of each treatment. CK means well-watered control plants without ethephon application. Drought means drought-stressed plants without ethephon application. ETH_D means drought-stressed plants with ethephon pre-treatment.

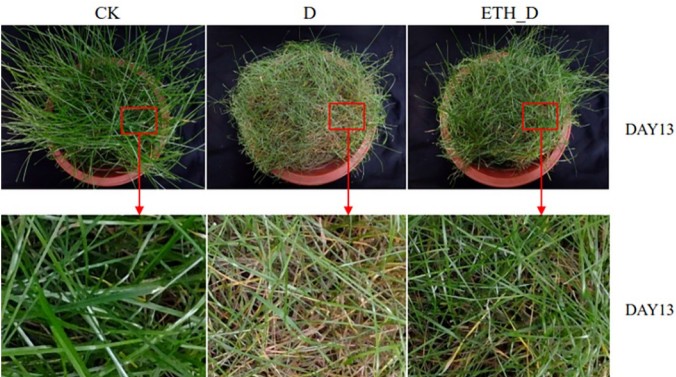

**Fig 3. Effects of ethephon on Kentucky bluegrass under drought.** CK means well-watered control plants without ethephon application. Drought means drought-stressed plants without ethephon pre-treatment. ETH_D means drought-stressed plants with ethephon pre-treatment.

protein databases, results showed that among the NR BLASTx best hits, Kentucky bluegrass unigenes were significantly similar to *Brachypodium distachyon* proteins (14,815, 19.1%), followed by *Aegilops tauschii* (14,710, 18.9%), *Hordeum vulgare* (10,517, 13.5%), *Triticum urartu* (7,283, 9.4%) and *Oryza sativa* (5,988, 7.7%) (Fig 4).

### 3.3 Differential expression and gene ontology

A total of 24,643 transcripts were identified as DEGs in one or more of the pairwise comparisons (Fig 5A). A large change of the transcriptome occurred in Kentucky bluegrass in response to drought stress (Fig 5A). A relative smaller change of transcriptome occurred in Kentucky blue grass in response to drought due to ETH pre-treatment (Fig 5A). The heatmap also indicates the overall effect of drought stress on transcription and allows for visualization of how ETH moderated the effects of drought stress on the transcriptome (Fig 6).

A total of 24,465 genes were either up- or down-regulated when comparing drought stressed to well-watered control plants (Fig 5A). Gene ontology (GO) and enrichment analysis identified 2877 biological processes, 1422 molecular functions, and 622 cellular components (Fig 7A). A total of 3,890 genes were either up- or down-regulated when comparing ETH

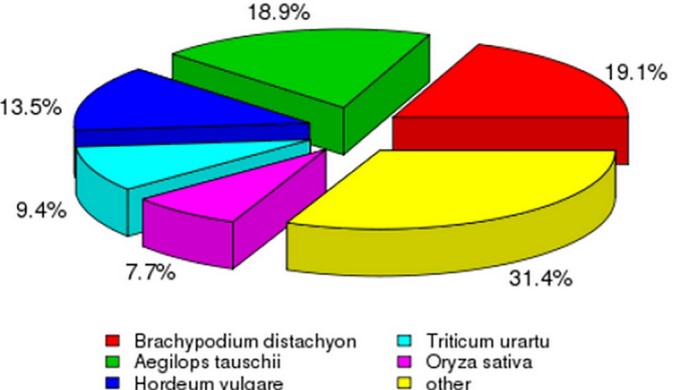

**Fig 4. Summary and taxonomic source of BLASTx matches for Kentucky bluegrass unigenes.** Percentage of unique best BLASTx matches of unigenes grouped by genus.

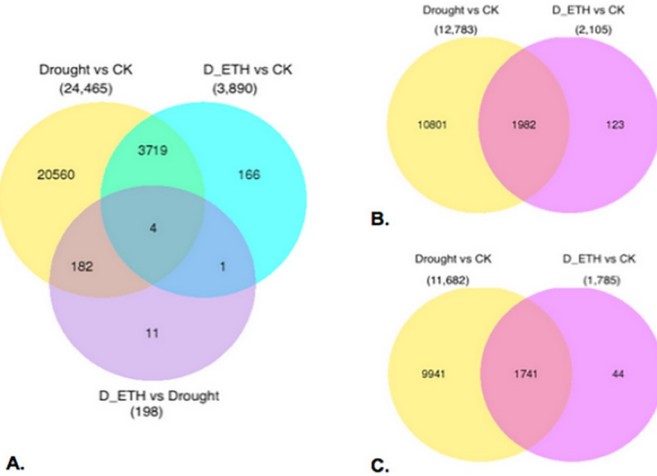

**Fig 5.** A. Venn diagram for all differentially expressed genes (DEGs) in Kentucky bluegrass B. Venn diagram for up-regulated genes in Kentucky bluegrass C. Venn diagram for down-regulated genes in Kentucky bluegrass. 'CK' means well-watered control plants without ethephon pre-treatment. 'Drought' means drought-stressed plants without ethephon application. 'ETH_D' means drought stressed plants with ethephon pre-treatment. DEGs were quantified at false discovery rate threshold (FDR) of 0.001 and log2 (fold change) larger than 2. Total DEGs for each comparison are shown in parenthesis.

primed drought stressed to well-watered control plants (Fig 5A). Gene ontology (GO) and enrichment analysis identified 1892 biological processes, 863 molecular functions, and 404 cellular components (Fig 7B).

It seemed that ETH treatment help plants dealing with drought by regulating muchlesser genes (3,890 vs 24,465) (Fig 5A). Therefore, only the genes most relevant to drought stress and ETH application are focused on the discussion part.

## 3.4 qRT-PCR validation of RNA-Seq results

Eight differentially expressed genes were randomly selected, including four genes from CK. (*c111268_g1*, *c145507_g1*, *c117236_g1*, *c119413_g2*) and four genes from ETH_D (*c145664_g1*, *c128115_g1*, *c135104_g1*, *c93924_g1*). Results showed that these genes used for qRT-PCR were all consistent with the RNA-Seq results (Pearson's r = 0.98, P <0.001, Fig 8) (see S5 Data for data used to calculate the qRT-PCR validation of RNA-Seq).

## 3.5 Differentially expressed genes due to drought and ETH

Drought caused extensive gene expression changes while drought and ETH caused less gene expression changes in Kentucky bluegrass plants, which indicated that ETH help plants coping with drought by mediating the regulation of fewer genes in response to drought. Therefore, in order to find the genes only regulated by ETH under drought stress, DEGs of these two comparisons [(ETH_D vs CK) vs (Drought vs CK)] were compared. Results showed 5.8% (123 out of 2105) of the transcripts were up-regulated (Fig 5B) and 2.5% (44 out of 1785) were down-regulated (Fig 5C) (DEGs up-regulated and down-regulated of [(ETH_D vs CK) vs (Drought vs CK)] were listed in S6 and S7 Data respectively). The mechanism of ethephon on drought tolerance of Kentucky bluegrass was analyzed by identifying DEGs involved in [(ETH_D vs CK) vs (Drought vs CK)] (Fig 9).

**3.5.1 Plant hormone, signal transduction and plant defense.** Ethylene Responsive Factor (AP2/ERF) family are conservatively widespread in the plant kingdom. Although the

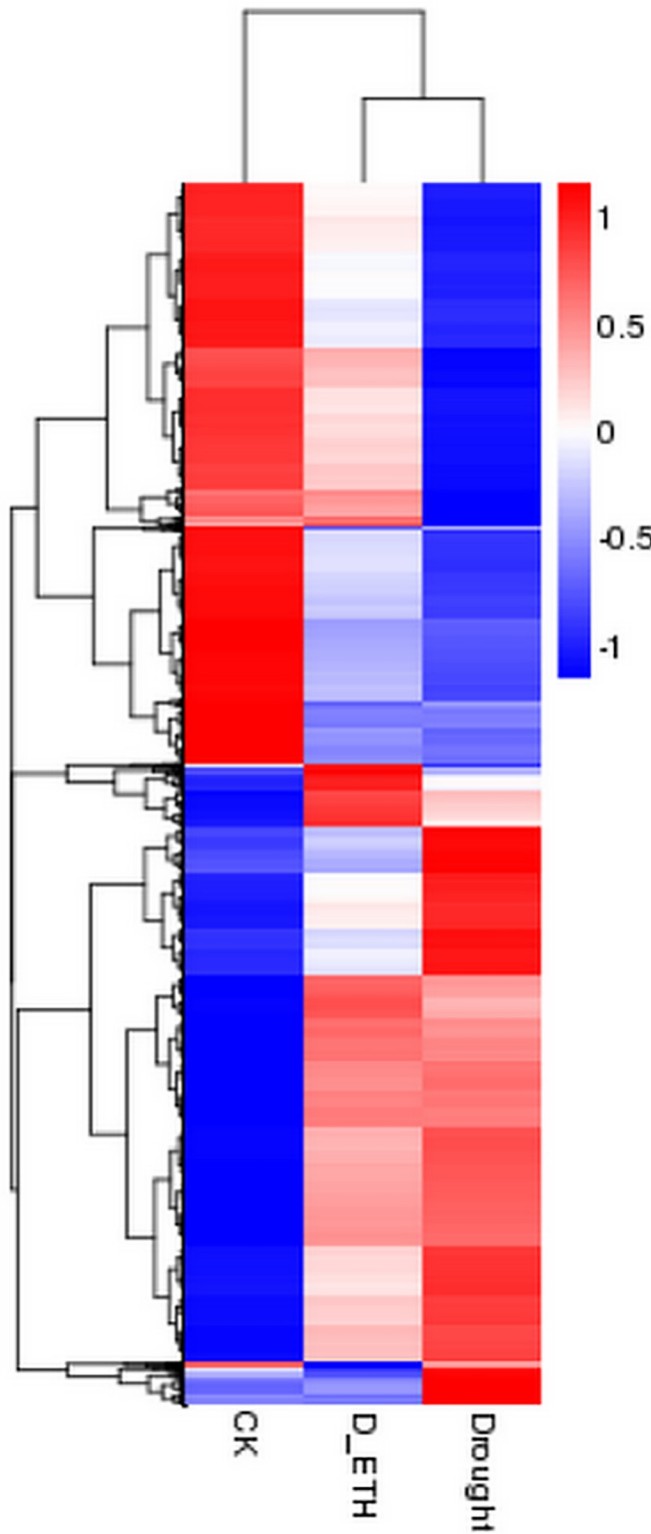

**Fig 6. Heat map of all differentially expressed genes in Kentucky bluegrass.** 'CK' means well-watered control plants without ethephon pre-treatment. 'Drought' means drought-stressed plants without ethephon application. 'ETH_D' means drought-stressed plants with ethephon pre-treatment.

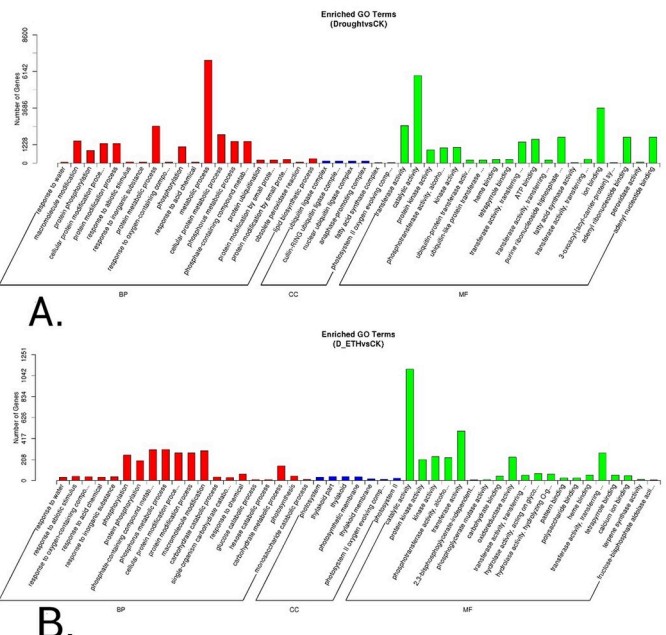

**Fig 7. Enriched GO terms.** A. Drought vs CK; B. ETH_D vs CK. 'CK' means well-watered control plants without ethephon pre-treatment. 'Drought' means drought-stressed plants without ethephon application. 'ETH_D' means drought-stressed plants with ethephon pre-treatment.

original acronym ERF, Ethylene-responsive transcription factor, has been maintained, responsiveness to the growth regulator ethylene is not a universal feature of this protein superfamily [37]. Two ERF genes were up regulated by ETH under drought in our study, *ERF113* (2.3 fold) and *ERF115* (2.4 fold). ERF113, also known as RELATED TO APETALA2.6L (RAP2.6L) in *Arabidopsis*, is induced by salt stress and drought [38, 39]. Additionally, *ERF113* transcription is responsive to JA, salicylic acid, ABA and ethylene [39]. Correspondingly, *ERF113* overexpression confers resistance to stresses that activate these hormones. For instance, overexpression of *ERF113* triggers stomatal closure and enhances waterlogging tolerance [40]. In addition to the response to hormonal cues, ERF113 activity can further be linked to

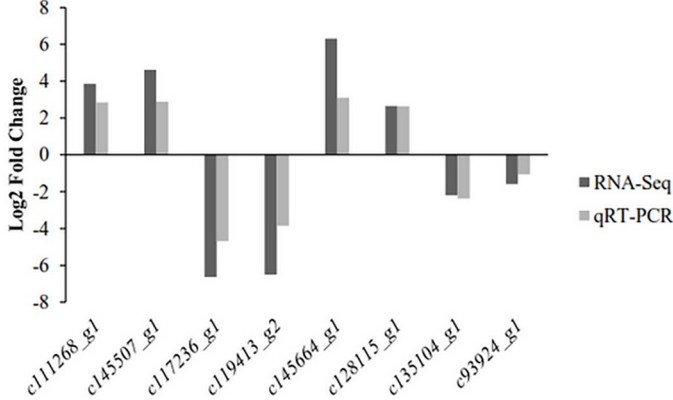

**Fig 8. Validation of DEGs data by qRT-PCR.** 'CK' means well-watered control plants without ethephon pretreatment. 'Drought' means drought-stressed plants without ethephon application. 'ETH_D' means drought-stressed plants with ethephon pre-treatment.

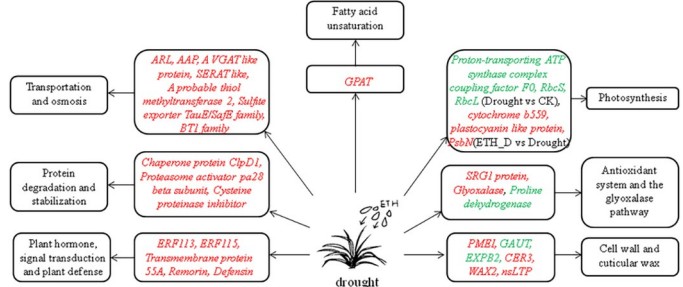

**Fig 9. The mechanism for how ethephon regulates drought response and improves drought tolerance of Kentucky bluegrass.** The gene in red indicates that the gene is up-regulated and the gene in green indicates that the gene is down-regulated.

developmental processes, such as shoot regeneration from root explants and ovule development [41]. ERF113 also has a role in promoting cell division that is induced by wounding [38, 42]. ERF115 drives the quiescent center (QC) cell division in a brassinosteroid-dependent way but is restrained through proteolysis by ubiquitin ligase. The QC plays an essential role during root development by creating a microenvironment that preserves the stem cell fate of its surrounding cells. Maintaining a stem cell subpopulation that is used to replace damaged stem cells might represent a general mechanism to maintain a functional stem cell niche under stress conditions [43].

It is well known that phosphatidylinositol 4,5-bisphosphate (PI(4,5)P2) plays important roles not only as a precursor lipid for generating second messengers but also as a regulator of cytoskeletal re-organization [44]. Recent examples of ion channel regulation by PI(4,5)P2 have been recently reported in plants. Since PI(4,5)P2 is mostly found in the plasma membrane, PI(4,5)P2 dependence is thought to restrict channel/transporter activity into this compartment and is important for stomatal opening [45]. As for the metabolizing pathways of PI(4,5)P2, there are three possible routes. One is conversion to PI(3,4,5)P3 by phosphatidylinositol 3-kinase. Second is hydrolysis to I(1,4,5)P3 and diacylglycerol by phospholipase C (PLC). Third is hydrolysis by PI(4,5)P2 phosphatase to PI(4)P [46]. Thus, PI(4,5)P2 levels are regulated by a balance of these metabolizing enzymes and synthesizing enzymes [47]. *Type II PI (4,5)P2 phosphatase* (*Transmembrane protein 55A*, 4.7 fold) is involved in the third PI(4,5)P2 metabolizing route: dephosphorylating the D4 position of PI(4,5)P2. Therefore, up regulation of this gene might decreased content of PI(4,5)P2 which may assist plant drought adaptation through stomata closure, ion channels activity and other transduction pathways involving second messengers derived from PI(4,5)P2 [48].

Transcript of a *remorin* gene (*c103095_g1*, Inf) was only detected in drought-stressed plants with ETH pre-treatment. This protein was named remorin due to its hydrophilic profile and its ability to attach to plasma membrane [49]. They probably facilitate cellular signal transduction by direct interaction with signaling proteins such as receptor-like kinases and may dynamically modulate their lateral segregation within plasma membranes [50]. The diverse and precise biological roles of different remorins remain to be investigated. However, the absence of remorins in algae, but their presence in mosses, ferns, and higher plants, suggests that the emergence of remorins coincided with the colonization of land and dealing with adverse drought and other osmotic stressed conditions [51, 52]. Transgenic *Arabidopsis* plants overexpressed *heterologous remorin* gene from mulberry [53] or foxtail millet (*Setaria italica*) [49] showed improved tolerance to abiotic stress including dehydration and salinity. How ETH treatment regulated *remorin* gene in response to drought is not clear. Yue et al. [49]

reported that there is an DRE core elements in the promoter region of foxtail millet *remorin gene 6* (*SiREM6*). One ABA responsive DREB transcription factor can bind to the DRE core elements. These results together suggest that ETH treatment might promote gene expression of *remorin* gene during drought stress in an ABA dependent signal transduction pathway.

Plant *defensin* (*c126749_g1*) gene is up regulated in ETH pre-treated Kentucky bluegrass plants under drought stressed conditions (2.7 fold). Plant defensins are small, highly stable, cysteine-rich peptides and they constitute an important part of the innate immune system primarily against fungal pathogens [54, 55]. In addition to their role in biotic response, plant defensin also has potential in inducing abiotic stress tolerance. Many reports revealed that plant *defensin* is also up regulated by salicylic acid, abscisic acid, ethephon and wounding [56–58]. Therefore, under drought stress conditions ethephon may up-regulate *defensin* expression to induce drought tolerance.

**3.5.2 Protein degradation and stabilization.** A few genes associated with protein degradation and stabilization were up-regulated by ETH and drought treatment. For instance, a gene encoding *chaperone protein ClpD1* (3.7 fold) was up-regulated in ETH treated plants under drought. ClpD1 may interact with a ClpP-like protease involved in degradation of denatured proteins in the chloroplast [59]. Previous studies revealed that ClpD1 plays a positive role during dehydration and salt stress [59, 60]. *Ubiquitin* (1.7817 fold) was also up-regulated by ETH under drought. The major function of ubiquitin is to facilitate protein degradation as an important component of the ubiquitin 26S proteasome system (UPS) in plant responses to abiotic stresses [61]. *Proteasome activator pa28 beta subunit* gene (1.8587 fold) was up regulated by ETH under drought. PA28 are activators that bind to proteasomes and stimulate the hydrolysis of peptides [62, 63]. Proteins of these up-regulated genes might perform an important role of removing potentially toxic proteins and misfolded or oxidized proteins that may accumulate as a result of exposure to drought stress. However, a *cysteine proteinase inhibitor* gene (1.74 fold) was up-regulated by ETH under drought. Cysteine proteases play an essential role in plant growth and development but also in senescence and programmed cell death [64]. They are among the plant proteases and are increased in their activity following stress [65, 66]. If the activity of the cysteine proteinases is too high, proteins required for metabolic processes degraded rapidly due to proteolysis [67]. It is therefore of great importance that the activity of the cysteine proteinases are accurately controlled in order to cope with drought. This is achieved, possibly through up regulation of *cysteine proteinase inhibitor*. Previous transgenic studies confirmed that *cysteine proteinase inhibitor* played active role in response to stress including drought [67–69].

**3.5.3 Transportation and osmosis.** ARFs confers tolerance to biotic and abiotic stresses in plant species [70]. Overexpression of an *adenosine diphosphate-ribosylation factor* gene from the halophytic grass *Spartina alterniflora* confers salinity and drought tolerance in transgenic *Arabidopsis* [71]. Ectopic expression of *ADP ribosylation factor 1 (SaARF1)* from smooth cordgrass (*Spartina alterniflora* Loisel) confers drought and salt tolerance in transgenic rice and *Arabidopsis* [72]. *ADP-ribosylation factor-like protein (ARL)* (Inf) (If the normalized read-counts of one particular gene in one sample is 0 and not 0 in another sample, fold change would be Inf or -Inf) belong to Ras superfamily of small GTP-binding proteins (GTPases). ARLs were identified on the basis of their sequence similarity with ARFs. GTP-binding has been shown for most ARL proteins, but all ARLs are essentially devoid of GTPase activity and activities described for ARF isotypes. Some ARLs appear to be involved in the regulation of protein and/or vesicle transport between cell organelles (ARL1, ARL4) or in the regulation of enzymatic activities controlling these processes (Arfrp1). In potato (*Solanum tuberosum*), six clones of *ADP-ribosylation factor-like protein* were up-regulated by salt treatment [73]. Gene expression induction of *ARL* in ETH treated plants under drought might facilitate the plants

for higher exchange rates of ions, proteins and other molecules by protein and/or vesicle transport pathway.

Amino acids are essential components of plant metabolism, not only as constituents of proteins, but also as precursors of important secondary metabolites and as carriers of organic nitrogen between the organs of the plant. Transport across intracellular membranes and translocation of amino acids within the plant are mediated by membrane amino acid transporters. However, the substrate selectivity and affinity of membrane amino acid transporters are generally different. Amino acid transport also plays a key role in leaf senescence and seed germination. Clearly, amino acid transport is a fundamental activity in plant growth [74]. *A putative amino acid permease* (AAP, 4.6 fold) was identified in ETH treated drought stressed plants. AAP is a family of amino acid transporters that preferentially transport glutamine, asparagine, glutamate, and neutral amino acids into plant cells [75]. GABA is a key regulator of ion channels in plants and animals [76]. Abiotic stresses including salt, anoxia, hypoxia, heat, mechanical damages, drought, cold, and waterlogging drive GABA accumulation in plants [77]. Vesicular GABA transporter (VGAT) belongs to solute carrier family 32 (vesicular inhibitory amino acid transporter) [78]. We identified a *VGAT like protein* (5.3 fold) regulated by ETH and drought in Kentucky bluegrass. The VGAT is known as the amino acid/auxin permease superfamily [79]. Two genes with low similarity to a vesicular GABA transporter, potentially functioning in cellular transport processes were also found to be commonly up-regulated in response to cellular water deficit in *Arabidopsis* [80]. It is possible that up-regulation of these amino acid transporters might be involved in amino acid-based osmotic regulation under drought in response to ETH treatment.

Sulfur plays a pivotal role in plant metabolism and development. Evidence is emerging that a number of non-protein and protein thiols, together with a network of sulphur-containing molecules and related compounds, also fundamentally contribute to plant stress tolerance [81]. *A serine acetyltransferase like protein* (*SERAT like*, 4.6 fold) and *a probable thiol methyltransferase 2* (2.6 fold) were up regulated by ETH and drought. Cysteine (Cys), as the first organic-reduced sulfur compound, contributes not only to life as building blocks in proteins, but it also serves as a precursor for the synthesis of Methionine (Met), glutathione (GSH), cofactors, essential vitamins, sulfur esters, and other sulfur derivatives. Cys synthesis is catalyzed by the sequential action of SERAT and O-acetylserine (thiol)lyase (OASTL), links Ser metabolism to Cys biosynthesis [82]. Overexpressing of *bacterial SERAT* in transgenic tobacco plants lead to increased resistance to oxidative stress [83]. *Sulfite exporter TauE/SafE family* gene (2.5 fold) were involved in regulation of plant-type hypersensitive response and they were defense-related and enriched with clock regulatory elements [84]. The proteins are involved in the transport of anions across the cytoplasmic membrane during taurine metabolism as an exporter of sulfoacetate [85]. *Sulfite exporter TauE/SafE* gene was also up-regulated in drought-stressed *P. euphratica* leaves [86].

*Biopterin transporter* (*BT1 family*, *transmembrane protein*, 4.1 fold, PFAM ID PF03092) belongs to the folate-biopterin transporter (FBT) family [87]. Folates take part in virtually every aspect of plant physiology. They play a role of donors and acceptors of one-carbon groups in one-carbon transfer reactions that take part in formation of numerous important biomolecules, such as nucleic acids, panthothenate (vitamin B5), amino acids [88]. The role of folates in plant stress response are also important. Folate supplementation was demonstrated to improve plant biotic stress resistance. Moreover, folate metabolism was shown to be differentially regulated in response to various abiotic stress conditions that pointed out its importance and possible specific adjustment in response to different stresses. Altogether these findings indicate that physiological roles and regulation of folate metabolism during development and stress response are important elements to be considered in the pursuit of crops with

better productivity and improved stress tolerance [88]. Folate/biopterin transporter gene was induced by 48-h rehydration and inhibited by drought stress in shoot and panicle of rice (*Oryza sativa*) [89]. BT1 is also induced by Nitro-Linolenic Acid which plays strong signaling role in the defense mechanism against different abiotic-stress situations in *Arabidopsis* [90].

**3.5.4 Antioxidant system and the glyoxalase pathway.**   Oxidative stress is one of the common consequences of abiotic stress including drought in plants, which is caused by excess generation of reactive oxygen species (ROS). *SRG1 protein* (*c127636_g2*, 2.3 fold), senescence-related gene, is a new member of the Fe(II)/ascorbate oxidase superfamily and SRG1 protein detoxify reactive oxygen produced during the oxidative stress induced by drought. It is revealed that *SRG1* homolog gene in potato putatively contributes to potato drought tolerance [91]. *SRG1* is regulated by WRKY transcription factors and involved in defense signaling pathways in *Arabidopsis* [92]. Therefore, up regulation of *SRG1 protein* by ETH might help improving ROS scavenging ability of Kentucky bluegrass under drought.

In line with ROS, plants also produce a high amount of methylglyoxal (MG) in response to various abiotic stresses, which is highly reactive and cytotoxic. MG and ROS accumulation results in an imbalance in different cellular metabolic processes. The glyoxalase pathway acts to control excessive accumulation of MG and ROS in the system, either directly or in cooperation with other pathways involved in stress response [93, 94]. In addition, transgenic approaches in various plant models also have demonstrated the ability of glyoxalases in imparting abiotic stress tolerance [95, 96]. Therefore, we propose that up regulation of *glyoxalase* (*c124305_g1*, 3.8 fold) by ETH might help Kentucky bluegrass plants detoxify MG and improve plants performance under drought.

Two *proline dehydrogenase unigene* (-2.3 and -2.6) catalyzes the first step in proline degradation and it is the rate-limiting enzyme in proline degradation [97]. Down regulation of *proline dehydrogenase* gene could lead to slower degradation of proline which would be an advantage [98].

**3.5.5 Cell wall and cuticular wax.**   During drought, it is important for plants that the cell wall is rigid enough to resist internal turgor pressure. The *plant invertase/pectin methyl esterase inhibitor* (*PMEI*, *c133760_g1*, 2.1 fold) inhibits demethylesterification of pectins by inhibition of endogenous PME, which keeps up highly methylated pectin [99]. Pectin, one of major components of the plant cell wall, has been shown to play a key role in modulating cell wall structure in response to drought stress [100, 101]. Degree of methylesterification of pectins related to interaction of PME and PMEI could affect mechanical properties of cell wall such as plasticity, extensibility, fluidity and thickening and those properties could enable adaptation and/or resistance to abiotic stress [102, 103]. In addition, pectin may play important roles in drought adaptation through modulating stomata movement [99, 104]. Our study suggested that inductions of *PMEI* expression provide beneficial effects in plants drought responses and this result was consistent with other studies [86, 105]. A reduced amount of pectin, coincided with an increase in firmness. *Putative galacturonosyltransferase (GAUT*, *c121058_g1*, -2.8 fold) are required for the synthesis of pectin [106, 107]. Expansins are cell wall proteins that are implicated in the control of cell extension via the disruption of hydrogen bonds between cellulose and matrix glucans. Since they function as cell wall-loosening proteins [108], down regulation of *expansin (EXPB2*, *c139601_g1*,-1.9 fold) by ETH may improve cell wall plasticity of plants during prolonged drought.

Cuticular wax has been implicated in defense mechanisms against biotic and abiotic stress including drought [109–111], possibly because the waterproof cuticular wax can counteract non-stomatal water loss during periodic drying and drought stress [112]. Two genes contributing to cuticular wax synthesis were identified in ETH and drought treated Kentucky bluegrass plants, *ECERIFERUM3* (*CER3*, *c90612_g1*, 2.5 fold) and *WAX2* (*c135869_g1*, Inf). CER3 is

important for cuticular wax synthesis [113]. WAX2 is involved in synthesis of leaf cuticular wax and also cutin composition [114, 115]. *One plant non-specific lipid-transfer protein (nsLTP, c120612_g1, 2.6 fold)* was also up regulated by ETH in Kentucky bluegrass under drought. Plant non-specific lipid-transfer protein form a protein family of small, basic proteins ubiquitously distributed throughout the plant kingdom [116]. The members of this family are located extracellularly, usually associated with plant cell walls, and possess a broad lipid-binding specificity [117]. Plenty of studies reported that *nsLTP* genes played important roles in plants' drought responses. For example, three *nsLTPs* genes are drought inducible in tomato [118] and one sugarcane (*Saccharum* hybrid complex) *NsLTPs* gene was up-regulated by PEG-simulated drought [119]. Over expression of *nsLTP* gene from *Setaria italic* in tobacco resulted in higher levels drought tolerance compared to wild type plants [120]. Similarly, enhanced drought tolerance of transgenic potato plants over-expressing *non-specific lipid transfer protein-1 (STnsLTP1)* was also observed [121]. While the mechanisms remain elucidated, one possible role of LTP in elevating drought tolerance is to promote cuticle deposition [117, 122].

**3.5.6 Fatty acid unsaturation.** *Glycerol-3-phosphate acyltransferase* (*GPAT*, 2.9 fold) catalyzes the transfer of an acyl group from an acyl donor to the sn-1 position of glycerol 3-phosphate. There are three types of GPAT in plant cells; they are localized in plastids (including chloroplasts), in the cytoplasm, and in mitochondria, respectively. Genetic engineering of the unsaturation of fatty acids has been achieved by manipulation of the cDNA for the GPAT found in chloroplasts and has allowed modification of the ability of tobacco to tolerate chilling temperatures [123]. Introduction of the cDNA for shape *Arabidopsis* glycerol-3-phosphate acyltransferase (GPAT) confers unsaturation of fatty acids and chilling tolerance of photosynthesis on rice [124]. Xu et al. [125] suggest that leaf dehydration tolerance and post-drought recovery in Kentucky bluegrass was associated with their ability to maintain relative higher proportion and level of unsaturated fatty acids. These studies together with ours suggested that higher expression of *GPAT* by ETH may lead to higher level of unsaturated fatty acids and therefore increased drought performance of ETH treated Kentucky bluegrass.

**3.5.7 Photosynthesis.** Photosynthesis is one of the key processes to be affected by water deficits [126]. *RbcS* gene was down regulated under both drought (-5.5 fold) and ETH treated drought (-3.2 fold) conditions while down regulation of *RbcL* (-3.1 fold) was only detected in drought treated plants. This indicates $CO_2$ assimilation in ETH treated plant might be less inhibited by drought. *Proton-transporting ATP synthase complex, coupling factor F0* (-3.2068 fold) was only down regulated in ETH treated plants under drought. ATP synthase activity is strictly related to photosynthesis because it transfers protons through the thylakoid membrane. Decrease expression of *ATP synthase complex coupling factor F0* may protect the photosynthetic apparatus from photo-damage by mediating non-photochemical quenching [127]. In addition, decreased ATP under low RWC impairs protein synthesis, through inadequate energy supply, but may increase some types of proteins, e.g. molecular chaperones, because their production is regulated in different ways.

Three up-regulated DEGs (ETH_D vs Drought) related to Photosystem II and electron transport were found in our study, *cytochrome b559, plastocyanin like protein and Photosystem II reaction centre N protein (PsbN). Cytochrome b559 [alpha (gene psbE) and beta (gene psbF) subunits* (2.9 fold)] is an essential component of photosystem II, catalyzing photosynthetic oxygen evolution [128]. Cytochrome b559 also plays a significant protective role for Photosystem II against photo inhibition during drought stress [129–132]. *Plastocyanin like protein* (3.1 fold) is involved in electron transport and it is responsive to drought both in barley and cassava [133, 134]. *PsbN* (2.8 fold) is required for hetero-dimerization of PSII reaction center in the stroma lamellae, and is required for early PSII assembly and repair [135, 136]. In summary,

ETH pre-treatment might help plants maintain higher $O_2$ evolution rate under drought and protect photosystem from photo-damages.

## 4. Conclusion

Ethephon could improve the drought tolerance of Kentucky bluegrass by elevating RWC and decreasing MDA under drought. On a whole transcriptome level, ethephon application affected genes associated with plant hormone, signal transduction pathway, plant defense, protein degradation and stabilization, transportation, osmosis, antioxidant system, the glyoxalase pathway, cell wall, cuticular wax, fatty acid unsaturation and photosynthesis of Kentucky bluegrass under drought stress. Genes mentioned in the discussion may be beneficial to better understand the mechanism of ethephon affecting plants stress responses.

## Supporting information

**S1 Data. Primer pairs for qRT-PCR.**
(XLSX)

**S2 Data. Summary of sequencing data quality.**
(XLS)

**S3 Data. Frequency distribution of splicing transcript length.**
(XLSX)

**S4 Data. Reads alignment efficiency.**
(XLSX)

**S5 Data. qRT-PCR validation of RNA-Seq results.**
(XLSX)

**S6 Data. Up-regulated DEGs annotation of (ETH_D vs CK) vs (Drought vs CK).**
(XLS)

**S7 Data. Down-regulated DEGs annotation of (ETH_D vs CK) vs (Drought vs CK).**
(XLSX)

## Acknowledgments

The authors wanted to express thanks to Yinan Sun working at Beijing Forestry University for drawing the Kentucky bluegrass in Fig 9.

## Author Contributions

**Conceptualization:** Jiahang Zhang, Lixin Xu, Liebao Han.

**Formal analysis:** Jiahang Zhang.

**Investigation:** Jiahang Zhang, Yanan Gao.

**Methodology:** Yanan Gao, Lixin Xu.

**Project administration:** Lixin Xu, Liebao Han.

**Supervision:** Lixin Xu, Liebao Han.

**Writing – original draft:** Jiahang Zhang, Lixin Xu.

**Writing – review & editing:** Jiahang Zhang, Lixin Xu.

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
