## [Decision Letter · Decision Letter 0]

1 Oct 2021

PONE-D-21-16354Transcriptome analysis of Kentucky bluegrass subject to drought and ethephon treatmentPLOS ONE

Dear Dr. Xu,

Thank you for submitting your manuscript to PLOS ONE. After careful consideration, we feel that it has merit but does not fully meet PLOS ONE’s publication criteria as it currently stands. Therefore, we invite you to submit a revised version of the manuscript that addresses the points raised during the review process.

We look forward to receiving your revised manuscript.

Kind regards,

Anil Kumar Singh, Ph.D.

Academic Editor

PLOS ONE

Journal Requirements:

“The authors wanted to express thanks to the National Natural Science Foundation of China for financial support (grant numbers 31971770).”

“the National Natural Science Foundation of China (grant numbers 31971770) (http://www.nsfc.gov.cn/)

3. PLOS requires an ORCID iD for the corresponding author in Editorial Manager on papers submitted after December 6th, 2016. Please ensure that you have an ORCID iD and that it is validated in Editorial Manager. To do this, go to ‘Update my Information’ (in the upper left-hand corner of the main menu), and click on the Fetch/Validate link next to the ORCID field. This will take you to the ORCID site and allow you to create a new iD or authenticate a pre-existing iD in Editorial Manager. Please see the following video for instructions on linking an ORCID iD to your Editorial Manager account: https://www.youtube.com/watch?v=_xcclfuvtxQ"

5. Please upload a new copy of Figure 5 as the detail is not clear. Please follow the link for more information: " ext-link-type="uri" xlink:type="simple">https://blogs.plos.org/plos/2019/06/looking-good-tips-for-creating-your-plos-figures-graphics/"
" ext-link-type="uri" xlink:type="simple">https://blogs.plos.org/plos/2019/06/looking-good-tips-for-creating-your-plos-figures-graphics/"

Reviewers' comments:

Reviewer's Responses to Questions

**Comments to the Author**

1. Is the manuscript technically sound, and do the data support the conclusions?

Reviewer #1: Yes

Reviewer #2: Partly

2. Has the statistical analysis been performed appropriately and rigorously? 

Reviewer #1: No

Reviewer #2: No

3. Have the authors made all data underlying the findings in their manuscript fully available?

Reviewer #1: Yes

Reviewer #2: Yes

4. Is the manuscript presented in an intelligible fashion and written in standard English?

Reviewer #1: No

Reviewer #2: Yes

5. Review Comments to the Author

Reviewer #1: In this manuscript, the authors have investigated how ethephon pretreatment could help to improve the drought tolerance of Kentucky bluegrass by carrying out a comparative transcriptomic studies on three different treatment groups i.e. control plants grown under well-watered conditions (CK), plants subjected to drought treatment without ethephon pretreatment (Drought) and plants subjected to drought treatment with ethephon pretreatment (ETH_D).

The conclusion appears to be that the ethephon pretreatment might improve the drought tolerance of Kentucky bluegrass by modulating the expression of several genes associated with protein degradation and stabilization, phytohormones, intracellular transport, antioxidant system and the glyoxalase pathway, cell wall and cuticular wax, signal transduction pathway, fatty acid unsaturation, photosynthesis and defense and osmoregulation.

The scientific content of the present study design is useful for understanding how ethephon pretreatment can help in improving the drought tolerance in Kentucky bluegrass. This study will also help to identify the potential candidate genes that can be genetically engineered to enhance drought tolerance in plants. However, the manuscript needs extensive revision for typographical errors and grammar.

Reviewer #2: The manuscript titled “Transcriptome analysis of Kentucky bluegrass subject to drought and ethephon treatment”, describe a technically sound piece of scientific research where Illumina RNAseq analysis has been performed on drought and ethephon treated Kentucky bluegrass and the relative water content and malondialdehyde content were measured, with validation of few differentially expressed genes. The authors made all data underlying the findings in their manuscript fully available. The language in submitted articles is clear, correct, and unambiguous. However, some major concerns need to be addressed

• Most conclusions are exaggerated and are drawn based on RNAseq data with qRT-PCR validation of randomly selected eight differentially expressed genes.

• The statistical analysis has not been performed for any of the experiments especially for Relative water content (RWC) and Malondialdehyde (MDA) content. Even though statement significantly increased and decreased is used without performing the statistical test. Based on MDA and RWC content, we cannot conclude that ethephon could improve the drought tolerance of Kentucky bluegrass. Few more assays need to be performed before drawing any of the conclusions.

• Authors need to show representative images of plants before and after drought stress with ethephon treatment, to understand any physiological changes in control, drought and ethephon treated plants.

• Authors have claimed that changes in the cell wall and cuticular wax along with proline, antioxidant enzymes and unsaturated fatty acids levels due to ethephon treatment has improved the Kentucky bluegrass drought tolerance, based on expression levels. Authors need to further support their findings and claims by performing assays and anatomical studies.

• Authors need to improve the quality of the figures provided. In Fig.6, provide the name of genes instead of gene I.D’s, as authors have not mentioned which gene I.D represent which gene. Also, need to discuss two downregulated genes (c117236 and c119413) in fig.6.

• The objective of this study was to investigate the effect of ethephon on drought tolerance of Kentucky bluegrass and to understand the underlying mechanism by analyzing and identifying genes involved in ethephon mediated drought tolerance improvement, which authors had not provide any mechanism or tentative figure which assimilates the whole study into hypothesis and randomly selected eight differentially expressed genes does not prove or show any mechanism as such. This study is a more theoretical approach than experimental in revealing the mechanism for how ethephon regulates drought response and improves drought tolerance of Kentucky bluegrass.

Overall, authors need to perform few more assays with proper statistical analysis which can support the conclusions drawn by authors and submit high-quality figures.

6. PLOS authors have the option to publish the peer review history of their article (what does this mean?). If published, this will include your full peer review and any attached files.

Reviewer #1: **Yes: **Monika Bhuria

Reviewer #2: No

---

## [Author Response · Author response to Decision Letter 0]

13 Nov 2021

Response to Reviewer #1: 

1. In the sub section 2.1, authors have discussed about the well-watered control plants with ethephon pre-treatment (ETH). However, authors did not provide any data related to this group.

Response: Discussions related to well-watered control plants with ethephon pre-treatment (ETH) was deleted. In this study, we want to investigate the effect of ethephon on drought tolerance of Kentucky bluegrass by comparing and analyzing DEGs between (ETH_D vs CK) and (Drought vs CK). Therefore, we decided to delete the corresponding part related to well-watered control plants with ethephon pre-treatment (ETH).

2. In the sub section 2.3, authors should provide manufacturer’s country for the TRIzol kit used for RNA isolation.

Response: The manufacturer’s country for the TRIzol kit used for RNA isolation has been added.

3. In the sub section 2.6, authors should provide manufacturer’s name and country for qPCR kit used for two-step qRT-PCR detection.

Response: The manufacturer’s name and country for qPCR kit used for two-step qRT-PCR detection have been added.

4. In the sub section 2.6, authors should mention the citation for evaluating the relative quantitative data using the 2-ΔΔ CT method.

Response: The citation for evaluating the relative quantitative data using the 2-ΔΔ CT method has been added.

5. In the sub-section 3.3, please properly mention the different sub-figures of Fig. 3 in the text while explaining the figure.

Response: Revised. 

6. In the sub-section 3.5, please correct the sentence by adding ‘regulation of’ in the line “with drought by mediating the regulation of fewer genes in response to drought”.

Response: Corrected.

7. In the sub-section 3.5.1, please mention the plant in which ERF113 have been characterized.

Response: Revised.

8. The manuscript needs to be proofread. Please correctly frame the sentences.

e.g. “Type II PI(4,5)P2 phosphatase (Transmembrane protein 55A, 4.7 fold) identified………..transduction pathways”. 

“Ras superfamily of small GTP-binding proteins (GTPases) the involvement of ARFs in …………….stresses in plant species”.

“Transgenic Arabidopsis plants overexpressing heterologous remorin gene………..including dehydration and salinity”.

“They are among the plant proteases that are increased in their activity following stress”.

“Abiotic stresses drive …………including salt, anoxia, hypoxia, heat, mechanical damages, drought, cold, and waterlogging”.

“During drought…………………..internal turgor pressure is important”.

“In addition, pectin may ………………by modulation of stomata movement”.

Response: We sincerely appreciated the carefulness and suggestions from the reviewer and revised accordingly.

9. The acronym for ADP-ribosylation factor-like protein is ARL and please correct the line as “ARF and ADP-ribosylation factor-like protein (ARL) belong to Ras”.

Response: We have corrected it as suggested.

10. Please italicize the scientific name of plants in the entire manuscript. e.g. “Spartina alterniflora Loisel” “Arabidopsis”, Oryza sativa

Response: Revised as suggested.

11. Proofread the entire manuscript for typographical and grammatical errors.

e.g. “Tanscript of a remorin gene…..” correct the spelling of Transcript.

“also serves as a precursor for the syntheses of Methionine (Met), glutathione” correct the spelling of synthesis.”

“Plastocyanin like protein (3.1 fold) is evolved in electron transport and…..” correct the spelling of involved.

“Plant defensins are small, highly stable, cysteine-rich peptides constitute a part of the innate…… pathogens”.

“ETH treatment caused up-regulation of genes under drought associated with protein degradation and stabilization.”

“Transport across intracellular membranes………………….mediated by membrane amino acid transporters generally differ in substrate selectivity and affinity”.

“The VGAT is initially……………….. know as the amino acid/auxin permease superfamily”.

“Sulfite exporter TauE/SafE gene were also up-regulated”

“While the mechanisms remains elucidated, one possible role….”

“Photosystem II and electron transport were found up regulated by drought in ETH treated plants……..”

“ETH treatment might help plants maintain higher O2 evolution…..”.

Response: We proofread the entire manuscript and revised accordingly. We really appreciated the reviewer for the time spent on this manuscript.

12. In the line, “Proline dehydrogenase (-2.3 and -2.6) catalyzes the”, Please mention what – 2.3 and – 2.6 are specifying and mention the gene name given in the RNA seq data. 

Response: Revised. Should be “Two proline dehydrogenase unigene (-2.3 and -2.6) catalyzes......”.

13. In the line, “CER3 (c90612_g1, 2.5 fold) and WAX2 (c135869_g1, Inf)” what does Inf specify?

Response: If the normalized readcounts of one particular gene in one sample is 0 and not 0 in another sample, fold change would be Inf or -Inf. 

14. Fig. 1A is not mentioned anywhere in the text.

Response: Fig 1A (Fig 1 in revised manuscript) was mentioned the sub section 2.1 for methods explanation.

15. Please mention the supplementary data where it is discussed in the text.

Response: We have mentioned supplementary data in the text where it has been discussed. 

16. In Fig. 6, perform the statistical analysis to determine the significance for the differences in the RNA seq and qRT-PCR.

Response: Instead of carrying out a statistical analysis by LSD value for the same gene between RNA-Seq results and qRT-PCR expression, we performed a correlation statistical analysis. The correlation between RNA-Seq results and qRT-PCR expression was analyzed to get Pearson’s correlation coefficients as describe in sub section 3.4 (Pearson’s r = 0.98 and P 0.001).

17. Author should follow authors guidelines of PLoS One for writing references.

Response: We have revised the references accordingly.

18. Authors should maintain uniformity while writing the references

Response: References are revised.

 

Response to Reviewer #2:

1. Most conclusions are exaggerated and are drawn based on RNAseq data with qRT-PCR validation of randomly selected eight differentially expressed genes.

Response: Conclusions from our study are drawn based on a verified RNAseq data set combined with analysis from various published data as discussed in the manuscript.

2. The statistical analysis has not been performed for any of the experiments especially for Relative water content (RWC) and Malondialdehyde (MDA) content. Even though statement significantly increased and decreased is used without performing the statistical test. Based on MDA and RWC content, we cannot conclude that ethephon could improve the drought tolerance of Kentucky bluegrass. Few more assays need to be performed before drawing any of the conclusions. 

Response: The statistical analysis result of Relative water content (RWC) and Malondialdehyde (MDA) content has been added and noted in the corresponding figure. As suggested, we provided additional images to support our physiological data. For Kentucky bluegrass, RWC and MDA are two key and confirmed indicators for drought tolerance and they are used and investigated in many turfgrass studies for drought tolerance comparison and evaluation. The relevant studied have been mentioned in the corresponding results and discussion.

3. Authors need to show representative images of plants before and after drought stress with ethephon treatment, to understand any physiological changes in control, drought and ethephon treated plants. 

Response: The representative images of Kentucky bluegrass plants in well-watered conditions, drought-stressed conditions and ethephon-pretreated plants under drought-stressed conditions have been added accordingly.

4. Authors have claimed that changes in the cell wall and cuticular wax along with proline, antioxidant enzymes and unsaturated fatty acids levels due to ethephon treatment has improved the Kentucky bluegrass drought tolerance, based on expression levels. Authors need to further support their findings and claims by performing assays and anatomical studies. 

Response: We agree that with corresponding assays and anatomical analysis, the conclusions of this study would be more consolidated and thoroughly confirmed. Meanwhile, this study could still provide meaningful gene regulation information to understand the mechanism underlying ethephon-promoted drought tolerance of Kentucky bluegrass. To date, there is few research on this particular aspect. We hope this study could lay a basic and informative foundation for more and more detailed and experimental research in the future.

5. Authors need to improve the quality of the figures provided. In Fig.6, provide the name of genes instead of gene I.D’s, as authors have not mentioned which gene I.D represent which gene. Also, need to discuss two downregulated genes (c117236 and c119413) in fig.6.

Response: Genes in Fig 6 (Fig 8 in the revised manuscript) were selected randomly to verify the effectiveness of RNA-Seq data for further expression level analysis. Genes that we discussed in the text were selected based on the comparison of DEGs between different sampling groups. Gene I.Ds and corresponding names were listed in the Supplementary Data 6 and 7, and were highlighted accordingly.

6. The objective of this study was to investigate the effect of ethephon on drought tolerance of Kentucky bluegrass and to understand the underlying mechanism by analyzing and identifying genes involved in ethephon mediated drought tolerance improvement, which authors had not provide any mechanism or tentative figure which assimilates the whole study into hypothesis and randomly selected eight differentially expressed genes does not prove or show any mechanism as such. This study is a more theoretical approach than experimental in revealing the mechanism for how ethephon regulates drought response and improves drought tolerance of Kentucky bluegrass.

Response: We appreciated the reviewer’s suggestion. The idea to generalize the tentative mechanism into figure is really helpful in improving our manuscript. As suggested, Fig 9 was added to provide a mechanism hypothesis based on results from the whole study.

---

## [Decision Letter · Decision Letter 1]

3 Dec 2021

Transcriptome analysis of Kentucky bluegrass subject to drought and ethephon treatment

PONE-D-21-16354R1

Dear Dr. Xu,

We’re pleased to inform you that your manuscript has been judged scientifically suitable for publication and will be formally accepted for publication once it meets all outstanding technical requirements.

Kind regards,

Anil Kumar Singh, Ph.D.

Academic Editor

PLOS ONE

Additional Editor Comments (optional):

Reviewers' comments:

Reviewer's Responses to Questions

**Comments to the Author**

1. If the authors have adequately addressed your comments raised in a previous round of review and you feel that this manuscript is now acceptable for publication, you may indicate that here to bypass the “Comments to the Author” section, enter your conflict of interest statement in the “Confidential to Editor” section, and submit your "Accept" recommendation.

Reviewer #1: All comments have been addressed

2. Is the manuscript technically sound, and do the data support the conclusions?

Reviewer #1: Yes

3. Has the statistical analysis been performed appropriately and rigorously? 

Reviewer #1: Yes

4. Have the authors made all data underlying the findings in their manuscript fully available?

Reviewer #1: Yes

5. Is the manuscript presented in an intelligible fashion and written in standard English?

Reviewer #1: Yes

6. Review Comments to the Author

Reviewer #1: Authors have substantially revised the manuscript and addressed all the raised concerns. The manuscript can be accepted in its current form.

7. PLOS authors have the option to publish the peer review history of their article (what does this mean?). If published, this will include your full peer review and any attached files.

Reviewer #1: **Yes: **Monika Bhuria

---

## [Editor Report · Acceptance letter]

7 Dec 2021

PONE-D-21-16354R1 

Transcriptome analysis of Kentucky bluegrass subject to drought and ethephon treatment 

Dear Dr. Xu:

I'm pleased to inform you that your manuscript has been deemed suitable for publication in PLOS ONE. Congratulations! Your manuscript is now with our production department. 

Kind regards, 

on behalf of

Dr. Anil Kumar Singh 

Academic Editor

PLOS ONE